# Cross-property deep transfer learning framework for enhanced predictive analytics on small materials data

Vishu Gupta[1], Kamal Choudhary [2,3], Francesca Tavazza [2], Carelyn Campbell [2], Wei-keng Liao[1], Alok Choudhary[1] & Ankit Agrawal [1✉]

Artificial intelligence (AI) and machine learning (ML) have been increasingly used in materials science to build predictive models and accelerate discovery. For selected properties, availability of large databases has also facilitated application of deep learning (DL) and transfer learning (TL). However, unavailability of large datasets for a majority of properties prohibits widespread application of DL/TL. We present a cross-property deep-transfer-learning framework that leverages models trained on large datasets to build models on small datasets of different properties. We test the proposed framework on 39 computational and two experimental datasets and find that the TL models with only elemental fractions as input outperform ML/DL models trained from scratch even when they are allowed to use physical attributes as input, for 27/39 ($\approx$ 69%) computational and both the experimental datasets. We believe that the proposed framework can be widely useful to tackle the small data challenge in applying AI/ML in materials science.

[1] Department of Electrical and Computer Engineering, Northwestern University, Evanston, IL 60208, USA. [2] Materials Measurement Laboratory, National Institute of Standards and Technology, Gaithersburg, MD 20899, USA. [3] Theiss Research, La Jolla, CA 92037, USA. ✉email: ankitag@eecs.northwestern.edu

The field of materials science has seen a growing application of artificial intelligence (AI) and machine learning (ML) techniques, which has significantly contributed to enhanced property prediction models as well as accelerated materials exploration and discovery[1–18]. In particular, the so-called deep learning (DL) algorithms – which are ML algorithms based on deep neural networks – have shown a remarkable capability to automatically and efficiently extract features from raw inputs and build accurate models for different properties of materials, often surpassing traditional ML techniques. This has been made possible due to the increasing availability of large materials databases[19–23], mostly based on simulations such as density functional theory (DFT)[24], e.g., Open Quantum Materials Database (OQMD)[19], Automatic Flow of Materials Discovery Library (AFLOWLIB)[20], Materials Project (MP)[21], and Joint Automated Repository for Various Integrated Simulations (JARVIS)[22]. Since the size of training data has a significant impact on the quality of ML/DL models[25], such large datasets have greatly catalyzed the development of a data-driven paradigm in materials science, popularly known as materials informatics[8,26–30].

However, highly accurate models are still limited to a few selected materials properties due to the non-uniform size of available data[31]. In other words, although the size of materials datasets is increasing, it is limited to a few properties that are relatively easy to measure/compute, and the field of materials science as a whole is still in the small data regime for the most part. There have been several studies where a small amount of data (<1000 samples) was used to train ML models[32–38]. However, these studies emphasize more on the importance of manual or domain knowledge-based feature engineering for training models for a specific materials property and less on the generalizability of the solution for the small data problem in materials science. Therefore, transfer learning (TL)[39] is often applied to tackle limited dataset problems by utilizing the rich features extracted from large datasets[40–56]. This involves first, training a model on a sufficiently large dataset, which serves as the starting point for the model training on the smaller dataset. For example, Jha et al.[45] used TL to transfer knowledge from a DL model (ElemNet) built on a large source dataset (OQMD) of DFT-calculated formation energies to small target datasets of DFT-based and experimental formation energies, i.e., same property as the source dataset. However, the unavailability of large datasets for many other properties such as exfoliation energy, dielectric constants, etc., greatly prohibits the application of DL and TL techniques in such cases.

In this work, we present a cross-property deep-transfer-learning framework that can transfer the knowledge learned by predictive models trained on large datasets to build predictive models on smaller datasets of other target properties, building upon previous works on transfer learning. The primary advantage of the proposed cross-property deep-transfer-learning framework is that it allows developing robust and accurate models on small datasets of properties for which other larger datasets may not be readily available. The block diagram of the proposed cross-property TL methodology is shown in Fig. 1 and consists of two steps. First, a DL model is trained from scratch on a big materials dataset (called the source dataset) of an available property (called the source property). The resulting model is called the source model. In the second step, the source model is used to build the target model of target property on a small target dataset. This can be done in two ways: (a) Fine-tuning the source model on the target dataset; and (b) Using the source model as a feature extractor to extract robust features for the target dataset, which can subsequently be used to build the target model using ML/DL. In this work, we use ElemNet[31] as the source model architecture, since it uses only raw elemental fractions (EF) as input, and has been shown to learn chemistry to materials using powerful DL techniques that have consistently shown to excel on raw inputs. Therefore, all the proposed cross-property TL models in this work are composition-based and require only EF as input. This not only simplifies the application of TL for the different properties used in this work, thereby demonstrating good generalization, but is also expected to facilitate the development of such models on other datasets and properties in the future. We compare the proposed cross-property TL models with scratch (SC) models, which perform model training from scratch with EF as input, as well as composition-based domain-knowledge-driven physical attributes (PA) as input for a more stringent comparison. The improvements and insights gained by using the proposed cross-property TL framework is expected to be useful for materials science researchers and practitioners to more gainfully utilize AI/ML/DL/TL techniques to overcome the small data challenge in materials science.

## Results

**Datasets.** We use two datasets of DFT-computed properties in this work: Open Quantum Materials Database (OQMD)[19] and Joint Automated Repository for Various Integrated Simulations (JARVIS)[22].

The dataset from OQMD comprises 341443 unique compositions, with their DFT-computed materials properties comprising formation energy, bandgap, stability, energy per atom, volume, and magnetic moment, as of May 2018. This dataset is used to derive the source dataset to train for the source model. JARVIS dataset comprises 28,171 unique compounds with up to 36 different properties as of July 2020 (https://ndownloader.figshare.com/files/22471022). We use two pre-processing steps to remove duplicate and overlapping compositions. First, to deal with duplicates arising due to different structures of the same composition, we only keep the most stable structure available in the database i.e. each data entry corresponds to the lowest formation energy among all compounds with the same composition, representing its most stable crystal structure. This is done independently for both datasets. Next, the common compositions between OQMD and JARVIS were removed from the OQMD dataset (denoted by OQMD-JARVIS) to avoid any overlap between the source and target datasets, which reduced the size of the source dataset from 341,443 to 321,140. This is to ensure that the pre-trained model does not see any compounds from the test set (which would be a subset of the target dataset). Further, each of the materials properties of the JARVIS dataset were confined to a range of permissible values, determined by looking at the values' distribution and consulting with domain experts. The data size for each materials property in the source dataset (OQMD and OQMD-JARVIS) and the target dataset (JARVIS) are shown in Supplementary Table 1 and Supplementary Table 2, respectively. Here, we use the OQMD dataset to refine the model architecture design, the OQMD-JARVIS dataset to perform the source model training used for TL-based models, and the JARVIS dataset to perform the target model training followed by materials property prediction and evaluation.

Modifications are made to the target dataset's materials properties to suit the DNN's model input which are shown in Supplementary Table 3. Total energy is excluded from the target materials properties as only differences of raw total energy are meaningful. Source model evaluation on the OQMD-JARVIS dataset uses only training and validation set with a random train:validation split of 90:10 to maximize the data used to train the source model for TL. Evaluation using the JARVIS dataset uses a holdout test set and a random train:validation:test split of 81:9:10.

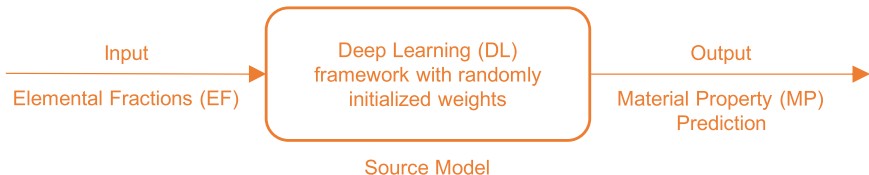

Step 1. Training from Scratch (SC) on (large) source dataset (e.g., OQMD)

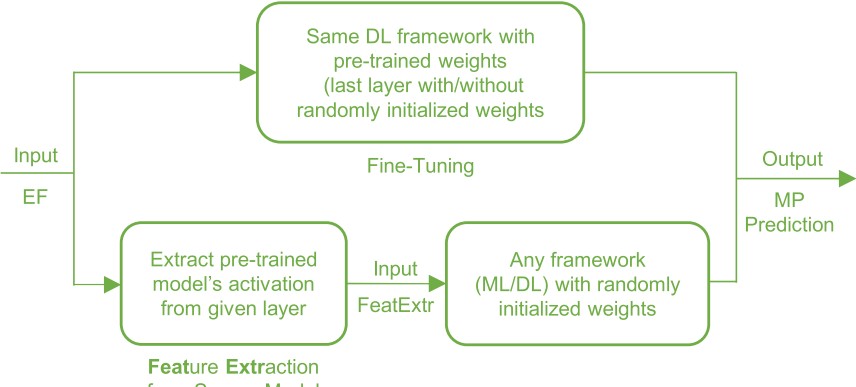

Step 2. Transfer Learning (TL) on (smaller) target dataset (e.g., JARVIS)

**Fig. 1 The proposed cross-property deep-transfer-learning approach.** First, a deep neural network framework (e.g., ElemNet) is trained from scratch, with randomly initialized network weights, on a big DFT-computed source data set (e.g., OQMD) using elemental fractions as input. Here, we refer to the model trained on the source dataset as the source model. Next, the same architecture (ElemNet) is trained on smaller target datasets (e.g., JARVIS) with different properties, using cross-property transfer learning, which can be done in two ways: 1. model parameters are initialized with that of the source model and then fine-tuned using the corresponding target dataset; or 2. source model is used to extract features for the target dataset in terms of activations from each layer of the source model, which are used as the input to build new ML and DL models (also called freezing method, if the features are extracted only from the last layer).

**Model architecture design**. We use ElemNet[31] as our base architecture for training the source models and perform transfer learning, as we use elemental fractions as input. ElemNet is a 17-layer architecture composed of fully connected layers[57], dropouts[58] at specific intervals, Adam[59] as the optimization function, and ReLU[60] as the activation function. It performs materials property prediction using an 86-dimensional vector of elemental fractions as the materials representation for the model's input. We provide a detailed explanation for all the model architectures used in this work in the Methods section. To improve the existing ElemNet model's performance, we made some changes to the framework and the architecture, and evaluated how it affects its accuracy of predicting formation energy using training data from OQMD. The original ElemNet model is written in TensorFlow 1 (TF1)[61]. For this work, we re-implemented ElemNet in TensorFlow 2 (TF2) with Keras[62] integration to take advantage of the improvements and high-level implementation of various components. This is also expected to facilitate code interpretability and reuse by domain scientists who may not be adept in programming. This change reduced the mean absolute error (MAE) from 0.0417 eV/atom for TF1 to 0.0405 eV/ atom for TF2. Second, the ElemNet model uses the monte carlo (MC) dropout[63] technique, where dropout is used during training as well as validation and testing phases. Using the dropout in this way produces different activations and outputs for the same input data each time we run the model, which may be useful for uncertainty quantification but does not extract consistent features for a given input. We thus disable dropout in this work so as to produce consistent and reproducible feature representations for a given input. We also found that disabling dropout further reduced the MAE from 0.0405 eV/atom to 0.0373 eV/atom. Thus, these two modifications helped us reduce the formation energy's MAE by ≈10% using the same 17-layer ElemNet architecture. We use this improved ElemNet

architecture to perform the model training for both the source and target dataset. The notations for specific models used in this work are described in Table 1. For multi-target model training for each source property, we first perform one run for all the SC/TL models using the same training-validation split. We then identify the best-performing model for each category in Table 1 and perform 10-fold cross-validation (CV) for that model. This is done independently for each source property for the TL models. The mean validation MAE from 10-fold CV is used to select the best performing modeling configuration for SC as well as TL. Since the selected modeling configurations have 10 models each (from 10-fold CV), the one with the least validation error is used to perform model testing on the holdout test set.

**Multi target transfer learning**. Here, we demonstrate the performance of TL models on different target materials properties when the source model is trained on formation energy as the source property for TL. We train ElemNet by using only elemental fractions with no explicit use of physical attributes as model input. Here the MAE obtained from the source model's training on OQMD-JARVIS is 0.0369 eV/atom, which is slightly different (in this case better) than the MAE obtained by performing the model training on the entire OQMD which was 0.0373 eV/atom. This pre-trained source model is then used to perform different types of transfer learning, including feature extraction, fine-tuning, and freezing (which is a feature extraction TL method if we perform feature extraction from the last layer of the source model). We compare the performance of TL models with the scratch (SC) models, i.e., ElemNet and traditional ML algorithms trained directly on the target dataset from scratch. The traditional ML algorithms used here include: Ada Boost, Elastic Net, Linear Regression, SGD regression, Ridge, Support Vector Machine, K-Neighbors, Decision Tree, Extra Tree, Bagging, Lasso,

**Table 1 Notations for the different scratch (SC) and transfer learning (TL) modeling configurations used in this work.**

| Notation | Description |
|---|---|
| Base | Naive model that simply uses the average property value of the training data as the predicted value |
| SC : ML(EF) | ML model trained from scratch using elemental fractions (EF) as input |
| SC : ML(PA) | ML model trained from scratch using physical attributes (PA) as input |
| SC : DL(EF) | DL model trained from scratch using EF as input |
| SC : DL(PA) | DL model trained from scratch using PA as input |
| TL : ML(FeatExtr) | ML model trained from the activations extracted from the source model (except for last layer) |
| TL : DL(FeatExtr) | DL model trained from the activations extracted from the source model (except for last layer) |
| TL : FineTune | Fine-tuning on the same DL framework using the pre-trained weights of source model |
| TL : ModFineTune | Fine-tuning on the same DL framework using the pre-trained weights of source model except for the last layer which has randomly initialized weights |
| TL : freezing | DL model trained from the activations extracted from the last layer of the source model |

and Random Forest, with hyperparameter tuning using extensive grid search for each of these ML algorithms as shown in Supplementary Table 4. Table 2 presents the prediction accuracy of the best SC and best TL model (selected based on their validation MAEs as shown in Supplementary Table 5) on the test set for each of the 39 target properties. Only formation energy is used as the source property for this analysis. Table 2 indicates that TL models with formation energy as the source property outperforms the SC models in 38/39 cases, i.e., in ≈97% of the cases. From Supplementary Table 5, we also find that within the TL models, feature extraction (FeatExtr) performs the best in more than half of the cases, and freezing performs the worst in all the cases.

The results clearly illustrate the benefit of using cross-property TL, i.e., using TL even when the materials properties of the source datasets and target datasets are different. We believe this is because the source model, if trained properly, can learn to extract useful features during the model training on the large datasets. However, not all source properties are expected to be the same with respect to the accuracy of the resulting TL models, which is what we investigate next.

**Multi source transfer learning**. Here, we look into the effect of using different ElemNet-based source models obtained by training on different source properties. Previously, we performed the model training for the source model on OQMD-JARVIS using only formation energy as the materials property. Other available materials properties in OQMD include band gap, volume, total energy per atom, stability, and magnetic moment. Supplementary Table 6 shows the best MAE obtained after training ElemNet on each of these materials properties. This gives us 6 source properties and 39 target properties, thereby 6 × 39 different combinations of TL scenarios to study the effect of cross-property TL.

Table 3 presents the prediction accuracy of the best SC and best TL model (selected based on MAE values from 10-fold CV as shown in Supplementary Table 7; all six source properties used for TL) on the test set for each of the 39 target properties. Table 3 indicates that multi-source TL models outperform the SC models in 38/39 cases, i.e., in ≈97% of the cases. From Supplementary Table 8, we can see that among the TL models, fine-tuning based TL model performed the best for 8/38 target properties, and FeatExtr based TL model using ML and DL performed the best for 3/38 and 27/38 target properties respectively. Encut was the only target property for which the best SC model was found to perform better than the best TL model.

**Comparison with physical attributes informed scratch models**. So far, we have observed the advantages of cross property TL using different source models on a variety of materials properties even when the source property and target property are different, and that TL models typically outperform SC models, all of which use only EF-based inputs, which is the simplest form of composition-based input. However, it is possible to infuse domain knowledge into the model by using more sophisticated composition-based attributes. For example, MagPie[5] represents a given composition by calculating a set of 145 physical attributes using easily available periodic table information, which was shown to universally work well for creating ML models for a variety of materials properties. In this work, although we want to use only composition-based EF inputs for the TL models for reasons described earlier, it would nonetheless be interesting to see how they would compare against stronger scratch models built using composition-based physical attributes (PA), thereby providing a more stringent test of the potential of the proposed ElemNet-based cross-property TL models. Hence, we use MagPie-derived attributes as the model input for the PA-based scratch models to compare against the TL models with EF-based inputs. Other types of attributes can be used in ML modeling of material properties as well, like structure-based ones, but they will not be investigated in this work Table 4 presents the prediction accuracy of the best SC and best TL model (selected based on MAE values from 10-fold CV as shown in Supplementary Table 9; SC models now also allowed to use PA as input) on the test set for each of the 39 target properties. As we can observe from Table 4, the TL models still perform better than SC models in 27/39 cases, i.e., in ≈69% of the cases. We also analyzed the results to identify the best performing TL techniques and the source properties across the various target properties, as shown in Supplementary Table 10. From our analysis, we observe that, although all the source models worked well for different properties, the source model trained on formation energy performed the best on most (9/27 or ≈33%) of the target properties. The source model trained on total energy performed the second-best (8/27 or ≈30%). The source model trained on stability performed the third-best (5/27 or ≈19%). The source models trained on magnetic moment, band gap, and volume performed best for three (3/27 or ≈11%), two (2/27 or ≈7%), and none of the target properties making volume the least contributing source model. Hence, it is advisable to use a source model trained on formation energy to perform cross-property transfer learning. We can also see that among the TL models, FeatExtr based TL models performed the best for 21/27 target properties. We also observed that for a given target property, if the same/similar source property is available (in our case, this was applicable to five target properties: BgOptb, Deltae, Magoszi, Magout, and BgMbj), using that as the source property to perform TL is better, as observed for 4/5 such cases. Further, in 3 out of those 4 cases, TL performed using fine-tuning was better than FeatExtr based TL. It is quite encouraging to observe that the proposed EF-based TL models outperform even the PA-based scratch models for more than half the cases. The majority of properties for which PA-based scratch models perform better than TL are very complicated ones, in the sense that they depend on several material properties at once. For instance, the effective mass

**Table 2 Prediction performance benchmarking for the prediction task of "Multi Target Transfer Learning" on the test set.**

| Property | Data size | Base | MAE of best SC model | MAE of best TL model |
|---|---|---|---|---|
| KLU | 28,056 | 18.77 | 11.96 | 11.37 |
| KAA | 28,171 | 5.234 | 2.978 | 2.821 |
| BgOptb | 28,163 | 0.988 | 0.279 | 0.251 |
| Deltae | 28,155 | 0.850 | 0.135 | 0.120 |
| Encut | 28,108 | 246.25 | 76.99 | 83.09 |
| Ehull | 27,297 | 0.131 | 0.055 | 0.050 |
| Magoszi | 25,844 | 1.225 | 0.438 | 0.405 |
| Magout | 25,357 | 1.176 | 0.393 | 0.369 |
| Eps | 25,150 | 3.829 | 1.462 | 1.304 |
| PPF | 16,250 | 650.5 | 543.1 | 508.6 |
| NPF | 16,250 | 658.1 | 546.3 | 493.0 |
| Pem300k | 16,763 | 1.918 | 1.293 | 1.111 |
| Nem300k | 16,760 | 1.918 | 1.282 | 1.183 |
| PSB | 14,439 | 163.30 | 68.34 | 60.53 |
| NSB | 14,144 | 108.69 | 57.83 | 53.32 |
| Meps | 11,349 | 4.905 | 1.926 | 1.832 |
| MaxM | 10,963 | 285.32 | 72.66 | 65.69 |
| MinM | 10,930 | 40.89 | 24.85 | 23.51 |
| ETC11 | 10,839 | 81.66 | 37.35 | 34.03 |
| ETC12 | 10,759 | 44.96 | 19.05 | 17.15 |
| ETC13 | 10,846 | 42.54 | 15.65 | 13.90 |
| ETC22 | 10,832 | 84.06 | 36.99 | 32.13 |
| ETC33 | 10,856 | 84.12 | 38.93 | 33.89 |
| ETC44 | 9986 | 29.55 | 17.24 | 14.76 |
| ETC55 | 9755 | 26.61 | 14.90 | 11.71 |
| ETC66 | 9739 | 27.59 | 15.83 | 13.81 |
| BulkKV | 10,743 | 49.11 | 11.83 | 11.01 |
| ShearGV | 10,209 | 24.28 | 11.90 | 11.11 |
| BgMbj | 7296 | 1.911 | 0.555 | 0.508 |
| Spillage | 3866 | 0.501 | 0.379 | 0.371 |
| SLME | 3006 | 9.439 | 7.193 | 6.877 |
| MaxIrM | 2302 | 426.0 | 108.2 | 104.6 |
| MinIrM | 2268 | 66.16 | 49.90 | 47.14 |
| PMDiEl | 2126 | 5.757 | 3.221 | 3.070 |
| PMDi | 2126 | 6.977 | 3.931 | 3.761 |
| PMDilo | 2126 | 2.577 | 0.847 | 0.791 |
| PMEij | 1123 | 0.520 | 0.436 | 0.415 |
| PMDij | 689 | 46.47 | 24.43 | 22.32 |
| Exfoli | 557 | 62.93 | 59.37 | 48.11 |

Only formation energy was used as the source property for this analysis. The table shows the test MAE of the best model selected using Supplementary Table 5 (based on validation MAE) when run on the test set for each of the target materials properties. All the model inputs are based on EF.

**Table 3 Prediction performance benchmarking for the prediction task of "Multi Source Transfer Learning" on the test set.**

| Property | Data size | Base | MAE of best SC model | MAE of best TL model |
|---|---|---|---|---|
| KLU | 28,056 | 18.77 | 11.96 | 11.15 |
| KAA | 28,171 | 5.234 | 2.978 | 2.832 |
| BgOptb | 28,163 | 0.988 | 0.279 | 0.251 |
| Deltae | 28,155 | 0.850 | 0.135 | 0.125 |
| Encut | 28,108 | 246.25 | 76.99 | 82.26 |
| Ehull | 27,297 | 0.131 | 0.055 | 0.050 |
| Magoszi | 25,844 | 1.225 | 0.438 | 0.405 |
| Magout | 25,357 | 1.176 | 0.393 | 0.376 |
| Eps | 25,150 | 3.829 | 1.462 | 1.262 |
| PPF | 16,250 | 650.5 | 543.1 | 494.0 |
| NPF | 16,250 | 658.1 | 546.3 | 512.3 |
| Pem300k | 16,763 | 1.918 | 1.293 | 1.228 |
| Nem300k | 16,760 | 1.918 | 1.282 | 1.197 |
| PSB | 14,439 | 163.30 | 68.34 | 60.25 |
| NSB | 14,144 | 108.69 | 57.83 | 54.53 |
| Meps | 11,349 | 4.905 | 1.926 | 1.776 |
| MaxM | 10,963 | 285.32 | 72.66 | 65.69 |
| MinM | 10,930 | 40.89 | 24.85 | 23.51 |
| ETC11 | 10,839 | 81.66 | 37.35 | 33.19 |
| ETC12 | 10,759 | 44.96 | 19.05 | 17.23 |
| ETC13 | 10,846 | 42.54 | 15.65 | 13.93 |
| ETC22 | 10,832 | 84.06 | 36.99 | 31.35 |
| ETC33 | 10,856 | 84.12 | 38.93 | 34.22 |
| ETC44 | 9986 | 29.55 | 17.24 | 14.63 |
| ETC55 | 9755 | 26.61 | 14.90 | 11.74 |
| ETC66 | 9739 | 27.59 | 15.83 | 13.10 |
| BulkKV | 10,743 | 49.11 | 11.83 | 11.28 |
| ShearGV | 10,209 | 24.28 | 11.90 | 11.01 |
| BgMbj | 7296 | 1.911 | 0.555 | 0.534 |
| Spillage | 3866 | 0.501 | 0.379 | 0.373 |
| SLME | 3006 | 9.439 | 7.193 | 6.420 |
| MaxIrM | 2302 | 426.0 | 108.2 | 103.4 |
| MinIrM | 2268 | 66.16 | 49.90 | 44.45 |
| PMDiEl | 2126 | 5.757 | 3.221 | 2.715 |
| PMDi | 2126 | 6.977 | 3.931 | 3.561 |
| PMDilo | 2126 | 2.577 | 0.847 | 0.774 |
| PMEij | 1123 | 0.520 | 0.436 | 0.367 |
| PMDij | 689 | 46.47 | 24.43 | 23.46 |
| Exfoli | 557 | 62.93 | 59.37 | 51.56 |

The table shows the test MAE of the best model selected using Supplementary Table 7 (based on validation MAE) when run on the test set for each of the target materials properties. The selected modeling configurations are listed in Supplementary Table 8. All the model inputs are based on EF.

(PEM300K and NEM300K) condenses structural and electronic information into one quantity, while Seebeck coefficients (NSb, PSb) are related to thermoelectric effects. Their complexity is likely why more sophisticated attributes incorporating domain knowledge help in such cases, while it is remarkable how close EF-only TL models get to the PA-based SC ones even in those cases. This provides more credence to the efficacy of both the source ElemNet model in successfully and automatically capturing relevant domain knowledge from only raw EF, as well as of the proposed cross-property TL methodology in effectively and appropriately transferring that knowledge for building predictive models for a variety of target properties on small target datasets.

Based on the above findings, our general recommendation for cross-property TL for a given target property would be to use feature extraction based TL using formation energy as the source property. Further, if the target property is also available as a possible source property, one may also want to try using the corresponding source model with a fine-tuning based TL scheme.

**Transfer learning on experimental data**. Here, we demonstrate the performance of TL models on experimental datasets with formation energy and band gap as materials properties. As we observed from our previous analysis that for a given target property, using formation energy as the source property and/or the same source property to perform TL is better, we use the source model trained on formation energy and band gap for this analysis. All the model training and testing are performed with the same experimental setup and pre-processing with EF as model input for TL models and EF/PA as model input for SC models. For experimental band gap, as the corresponding formation energy values were not available, the duplicate compositions were handled by only keeping the entry with the lowest band gap. Since there are only two experimental datasets, here we performed a more rigorous 100-fold CV for both SC and TL models, in order to get more MAE values to facilitate more accurate statistical testing (see Supplementary Table 11), and report the performance of the best model (selected based on the

**Table 4 Prediction performance benchmarking for the prediction task of "Performance against Physical Attributes" on the test set.**

| Property | Data size | Base | MAE of best SC model | MAE of best TL model |
|---|---|---|---|---|
| KLU | 28,056 | 18.77 | **10.79** | 11.15 |
| KAA | 28,171 | 5.234 | **2.722** | 2.832 |
| BgOptb | 28,163 | 0.988 | 0.279 | **0.251** |
| Deltae | 28,155 | 0.850 | 0.135 | **0.125** |
| Encut | 28,108 | 246.25 | **76.99** | 82.26 |
| Ehull | 27,297 | 0.131 | 0.058 | **0.050** |
| Magoszi | 25,844 | 1.225 | 0.438 | **0.405** |
| Magout | 25,357 | 1.176 | 0.393 | **0.376** |
| Eps | 25,150 | 3.829 | 1.280 | **1.262** |
| PPF | 16,250 | 650.5 | 495.0 | **494.0** |
| NPF | 16,250 | 658.1 | **484.2** | 512.3 |
| Pem300k | 16,763 | 1.918 | **1.086** | 1.228 |
| Nem300k | 16,760 | 1.918 | **1.086** | 1.197 |
| PSB | 14,439 | 163.30 | **56.49** | 60.25 |
| NSB | 14,144 | 108.69 | **48.93** | 54.53 |
| Meps | 11,349 | 4.905 | 1.784 | **1.776** |
| MaxM | 10,963 | 285.32 | **57.38** | 65.69 |
| MinM | 10,930 | 40.89 | 24.12 | **23.51** |
| ETC11 | 10,839 | 81.66 | 34.53 | **33.19** |
| ETC12 | 10,759 | 44.96 | **16.60** | 17.23 |
| ETC13 | 10,846 | 42.54 | 14.09 | **13.93** |
| ETC22 | 10,832 | 84.06 | 34.62 | **31.35** |
| ETC33 | 10,856 | 84.12 | 35.82 | **34.22** |
| ETC44 | 9986 | 29.55 | 15.23 | **14.63** |
| ETC55 | 9755 | 26.61 | 12.40 | **11.74** |
| ETC66 | 9739 | 27.59 | 13.53 | **13.10** |
| BulkKV | 10,743 | 49.11 | 11.83 | **11.28** |
| ShearGV | 10,209 | 24.28 | 11.26 | **11.01** |
| BgMbj | 7296 | 1.911 | 0.555 | **0.534** |
| Spillage | 3866 | 0.501 | 0.410 | **0.373** |
| SLME | 3006 | 9.439 | 7.193 | **6.420** |
| MaxIrM | 2302 | 426.0 | **87.67** | 103.4 |
| MinIrM | 2268 | 66.16 | **38.52** | 44.45 |
| PMDiEl | 2126 | 5.757 | 3.911 | **2.715** |
| PMDi | 2126 | 6.977 | 4.336 | **3.561** |
| PMDiIo | 2126 | 2.577 | 0.847 | **0.774** |
| PMEij | 1123 | 0.520 | 0.436 | **0.367** |
| PMDij | 689 | 46.47 | 26.46 | **23.46** |
| Exfoli | 557 | 62.93 | 54.26 | **51.56** |

The table shows the test MAE of the best model selected using Supplementary Table 9 (based on validation MAEs from 100-fold CV) when run on the test set for each of the target materials properties. The selected modeling configurations are listed in Supplementary Table 10. The SC models are allowed to use PA as input as well, while TL models only use EF-based inputs. The lowest MAE values in each row are highlighted in bold.

validation MAEs from 100-fold CV) on the holdout test set for all the models. In addition, here, we also used an AutoML library called hyperopt sklearn[64] for the scratch ML models in addition to the extensive hyperparameter search to optimize hyperparameters for all the ML models.

Table 5 presents the prediction accuracy of the best SC and best TL model (selected based on MAE values from 100-fold CV as shown in Supplementary Table 11) on the test set for each of the two experimental target properties. As we can observe from Supplementary Table 11 and Table 5, the TL models outperform the SC models in both cases. For the experimental dataset on formation energy, TL:FineTune using formation energy as the source model and SC:DL(EF) performed the best for TL and SC respectively. For the experimental dataset on band gap, TL:DL(FeatExtr) using formation energy as the source model and SC:DL(PA) performed the best for TL and SC respectively. Interestingly, the proposed TL models were able to reduce the

MAE of the experimental dataset on formation energy to 0.0708 eV/atom as compared to the previous best recorded MAE of 0.0731 eV/atom from ref. [45] on the same test set. Another interesting observation from Supplementary Table 11 is that for the experimental dataset on band gap, TL model using feature extraction was more accurate than the TL model using fine-tuning. We believe this might be due to the relatively larger difference in the band gap values computed using DFT and measured via experiments, as it is well-known that band gaps are quite challenging to calculate accurately using DFT[65,66].

## Discussion

In this paper, we presented an AI/ML/DL framework for cross-property transfer learning by building various source models trained on materials properties from a large dataset, and transferring that knowledge to build better target models on various properties from smaller datasets for enhanced materials property prediction. To illustrate the benefit of the proposed approach, we built source models using a deep learning framework called ElemNet on the OQMD dataset for various materials properties present in the dataset. This trained model was then used to perform three kinds of transfer learning on the smaller JARVIS datasets to find that the proposed TL models work well even when the source property and target property are different, which is expected to be especially useful to build predictive models for properties for which big datasets are not available. We trained the DL model and traditional ML models from scratch to compare the transfer learning performance. The deep learning framework ElemNet was originally designed to predict formation energy using vector-based materials representation composed of 86 elemental fractions as the model input. We made careful changes in the model architecture design to improve the model's robustness and accuracy by modifying the software implementation framework and the architecture component's usage. We demonstrated the proposed approach's efficiency by evaluating and comparing the TL models with SC models with the same vector-based materials representation composed of 86 elemental fractions. We then performed a more stringent test of TL models by incorporating vector-based materials representation composed of 145 composition-derived physical attributes for only the SC models. We also perform additional statistical analysis by calculating the 95th percentile of the distribution of the absolute error ($Q_{95}$) for the best performing models of Tables 2–5, which is shown in Supplementary Table 12. Even when SC models are allowed PA as model input, TL models perform better than SC models in 26/41 cases for $Q_{95}$, i.e., in 63% of the cases.

In order to see if the observed improvement in accuracy of TL models over SC models is significant, we perform statistical testing to estimate the one-tailed $p$-value for comparing the test MAEs obtained on 41 target datasets (39 DFT-computed datasets and two experimental datasets) using the Signed Test[67,68] as we are dealing with different datasets with different properties, whose MAE differences may not be directly comparable[69]. Here, the null hypothesis is that "TL model is not better than SC model" and the alternate hypothesis is "TL model is better than SC model". If we take into account that the SC models are allowed PA as input, we get the $p$-value $= 0.00397$ for MAE comparison and $p$-value $= 0.04291$ for $Q_{95}$ comparison using a sign test calculator[70], thus rejecting the null hypothesis at $\alpha = 0.05$, suggesting that such a difference in test MAE/Q95 between SC and TL models is unlikely to have arisen by chance. We can thus infer that, in general, TL models perform significantly better than SC models for cross-property transfer learning. We also show the versatility of the best performing TL method used in our analysis (the feature-extraction TL method), which facilitates the extraction of

**Table 5 Prediction performance benchmarking for the prediction task of "Transfer Learning on Experimental Data" on the test set.**

| Property (unit) | Data size | Base | MAE of best SC model | MAE of best TL model |
|---|---|---|---|---|
| Formation energy (eV/atom) | 1643 | 1.0327 | 0.0964 | **0.0708** |
| Band gap (eV) | 4920 | 1.2061 | 0.4458 | **0.3551** |

The table shows the test MAE of the best model selected using Supplementary Table 11 (based on validation MAE) when run on the test set for each of the target materials properties. The SC models are allowed to use PA as input as well, while TL models only use EF-based inputs. The lowest MAE values in each row are highlighted in bold.

relevant and robust features on the target datasets that can then be used with any ML/DL technique, thereby providing flexibility and interoperability. Even within the extracted features, the features extracted from the first four layers were found to perform significantly better than the features extracted from later layers. The so called freezing technique, which is essentially feature extraction from the last layer, did not perform very well in our study. This might be because the representations can change significantly after every layer, gradually becoming more specific to the particular dataset used for source model training, thus making it difficult for the target DL model to learn the target property, especially for cross-property TL. Moreover, this seems to be consistent with the fact that of the 16 layers of ElemNet, the representations extracted from only the first four layers were found to result in accurate models. To see if there is a significant difference between the two fine-tuning techniques (FineTune and ModFineTune), we calculate the one-tailed $p$-value using Signed Test[67,68] as we have their results on 39 different datasets. Here, the null hypothesis is "ModFineTune is not better than FineTune" and the alternate hypothesis is "ModFineTune is better than FineTune". After comparing the results from 6 (source properties) × 39 (target properties) = 234 cases, we get the $p$-value = 0.00104 using a sign test calculator[70], thereby rejecting the null hypothesis at $\alpha = 0.05$, indicating that TL:ModFineTune performs significantly better as compared to TL:FineTune for cross-property transfer learning. These observations are intuitive and encouraging. Thus, when building a predictive model on a small target dataset with a given target property, if the same source property is available, it is advisable to try using the corresponding source model as well as formation energy as the source property, as we did for the experimental target datasets.

In order to understand how the performance of SC and TL models varies with the size of the training data on a fixed test set, we performed additional model training experiments for formation energy for different training data sizes using the same test set (10% of the total data size) to create a learning curve that shows prediction accuracy as a function of the training set size. Supplementary Fig. 1 shows that in general, TL models outperform SC models for all the training sizes for formation energy prediction. We also studied the correlation between the target properties, as shown in Supplementary Fig. 2. Although most of the materials properties that show pair-wise correlation can be explained with domain knowledge, we also observed some pair-wise correlations that are not directly explainable by domain knowledge. For example, we observe a strong positive correlation between band gap and PSb, Exfoli, and ETC55 and a strong negative correlation between PSb and Eps, PSb and Meps, band gap and Eps, band gap and Meps, PMDiIo and band gap, Deltae and band gap, SLME and band gap, KAA and band gap, PSb and KAA, PSb and NEM300K, and PSb and PEM300K. It would be interesting to see if it is possible to analyze and devise possible relations between strongly correlated materials properties from the dataset. Our analysis demonstrates that the prediction models benefit from leveraging the source model trained on a large dataset irrespective of the materials property used to perform the

source model training. The usefulness of TL methods such as feature extraction TL method and the large dataset's availability appear to benefit the prediction performance on small target datasets. This suggests that a pre-trained source model with a rich set of features learned on an extensive source dataset can be effectively used with the proposed cross-property TL framework to build enhanced models on small datasets. This is because they can better help the target model to learn their respective properties with more robustness and accuracy by refining the knowledge and rich set of hierarchical features learned by the source model. Although our current work only uses composition-based attributes to train source models and perform TL on target datasets for reasons described earlier, it is relevant to note that some of the recent deep learning models for predicting materials properties use graph-based methods[52,71–73] and some form of embedding with materials structure and/or composition-based information as input. One could thus try to build upon the proposed cross-property TL workflow by incorporating structure-based information into the workflow via vector-based structural attributes with fully connected deep neural networks[31,74,75] or graph-based neural networks[52,72,73], if structural information is available for both source and target datasets. The trained source models could then be used in a similar way to either extract robust features for the target datasets or directly fine-tune on the target datasets. The proposed framework is thus quite flexible and can combine different state-of-the-art deep learning models to improve upon the performance and potentially be applied to other materials properties for which enough data may not be available. Further, the proposed TL framework is expected to be easily adaptable to other scientific domains beyond materials science. The presented cross-property transfer learning technique is conceptually easy to implement, understand, use, and build upon. In the future, we plan to develop web-based tools deploying the best performing predictive models for different target properties, as well as extend the cross-property transfer learning framework along the lines discussed above.

## Methods

**Model architectures.** We illustrate the architecture of the deep learning framework ElemNet used for DL modeling in Supplementary Fig. 3. Due to the model input's numerical nature, ElemNet uses a series of fully connected layers[57] and ReLU[60] as the activation function. Dropout[58,63] is introduced at specific intervals to reduce the deep neural network's overfitting by regularizing the model. The technical modifications from the previous ElemNet model used in this work include re-implementing it in TensorFlow 2, and disabling the use of dropout in order to create uniform output features for the feature extraction TL method to maintain consistency, which also led to improvement in the accuracy of the model. As this deep learning framework only uses elemental fractions as the model input, it is called ElemNet. The implementation of the model used in this work is publicly available at ref. [76]. There are two broad types of representations that can be used for this type of problem: composition-based and structure-based, both of which have their advantages and limitations. By definition, the representations based on composition alone are unable to distinguish between different structure polymorphs of the same composition, which would end up being duplicates in the data, and thus would need to be removed before ML modeling. However, the resulting composition-based models could be used to make predictions without the need for structure as an input, which is useful since structure information is often unavailable or very expensive to calculate for new materials. Composition-based

models can thus be used effectively for virtual combinatorial screening to narrow down the vast space of promising materials systems by identifying promising compositions, which could then be further explored with methods for structure prediction and structure-based property prediction models. In this work, we use composition-based representation, similar to refs. [1,5,31,45,77,78]. To deal with duplicates arising due to different structures of the same composition, we only keep the most stable structure available in the database, i.e., the one with minimum formation energy, in line with previous works.

**Scratch and transfer learning models**. We implemented two basic types of scratch (SC) models and three basic types of transfer learning (TL) models in this work. For SC models, we perform the model training directly on the target dataset from scratch without providing the model with any form of pre-trained knowledge. The model architecture used for SC models includes traditional machine learning (ML) models, and deep learning (DL) framework called ElemNet. Two vector-based materials representation are used in this work: elemental fractions (EF), which is a vector of 86 elemental fractions corresponding to 86 elements; and 145 composition-derived physical attributes (PA). For transfer learning (TL) models, we use a model pre-trained on the source dataset using only EF as the vector-based-materials representation for the model input. The TL techniques incorporated in this work include Fine-Tuning, Freezing, and a novel feature extraction (FeatExtr) method. Fine-Tuning uses pre-trained model weights as a starting point for further training on the smaller dataset with the same architecture as the source model. As the property we want to perform TL on can be different for the source model and target model, we performed two types of fine-tuning. First, we performed the traditional or intuitive fine-tuning (FineTune), where we use all the source model weights without any modification. Next, we performed modified fine-tuning (ModFineTune), where instead of reusing the weights of the last layer, we randomly initialized those weights, since the last layer learns to predict the source property and might become over-specific to the source property, which could possibly make fine-tuning the model for the target property more challenging in some cases. FeatExtr uses the pre-trained model's activations from a given layer as the materials representation for each compound of the target dataset. As the DL model consists of 17-layers of architecture, we can have 16 different types of materials representation (excluding the last layer, which is the output layer) for each compound with varying sizes, as the materials representation size depends on the number of neurons in a given layer. For example, if we extract the materials representation from the first layer of the ElemNet model, each compound will be represented as a 1024-dimensional feature vector. The same can be done with other layers with different number of neurons. The resulting materials representation of the target dataset can thus be used as input to any ML/DL model. We thus performed the feature extraction TL technique both with traditional ML models as well as the ElemNet DL model. After training the model on features extracted from 16 different layers to be used as input to the ML models and the DL model, we report the best performing result. The freezing method is a special case of the feature extraction method where we use the pre-trained model's activation from the last layer used for FeatExtr, which is the layer before the output layer. The pre-trained DL model in this work is the ElemNet model trained on OQMD, and thus it implies no explicit use of PA, i.e., the model is trained by using only EF as input. We also incorporate a "Base" model as a naive baseline for comparison with other methods. This model always uses the average property value of all the training data provided to it as the predicted property of a test compound.

**Network and ML settings**. The deep learning framework, ElemNet, was implemented using Python and TensorFlow 2[61] and Keras[62]. The hyperparameters used in the ElemNet comprise of the following: ReLU as the activation function after each layer except for the last layer, Adam[59] as the optimizer with a mini-batch size of 32, learning rate as 0.0001. To prevent overfitting, we used early stopping with patience of 200 epochs to stop the model training if the performance did not improve on the validation set for 200 epochs. We implement the traditional ML models using Scikit-learn[79]. The traditional ML models used in this work are Ada Boost, Elastic Net, Linear Regression, SGD regression, Ridge, Support Vector Machine, K-Nearest Neighbors, Decision Tree, Extra Tree, Bagging, Lasso, and Random Forest. We carry out an extensive hyperparameter grid search to find the best hyperparameters for all the ML models which is shown in Supplementary Table 4. We use mean absolute error (MAE) as the loss function as well as the primary evaluation metric for all models. We performed all the modeling experiments using a fixed random seed of 1234567 to facilitate reproducibility.

## Data availability
The datasets used in this paper are publicly available from their corresponding websites-OQMD[19] (http://oqmd.org) and JARVIS[22] (https://jarvis.nist.gov), experimental formation energy (https://github.com/wolverton-research-group/qmpy/blob/master/qmpy/data/thermodata/ssub.dat), and experimental band gap from AutoMatminer[80] (https://github.com/hackingmaterials/automatminer). The preprocessed data used for training and testing the TL models in this work are available at (https://doi.org/10.5281/zenodo.5533023). Source data are provided with this paper.

## Code availability
The codes required to perform TL (using both feature extraction and fine-tuning) used in this study is available at https://doi.org/10.5281/zenodo.5533023.

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

## Acknowledgements

This work was performed under the following financial assistance award 70NANB19H005 from U.S. Department of Commerce, National Institute of Standards and Technology as part of the Center for Hierarchical Materials Design (CHiMaD). Partial support is also acknowledged from Department of Energy (DOE) awards DE-SC0014330, DE-SC0019358, and DE-SC0021399.

## Author contributions

V.G. designed and carried out the implementation and experiments for the cross-property deep transfer learning framework under the guidance of A.A., A.C., and W.L.; K.C., F.T., and C.C. provided the necessary domain expertize for this work. V.G., A.A., K.C., and F.T. wrote the manuscript. All authors discussed the results and reviewed the manuscript.

## Competing interests

The author declares no competing interests.

**Additional information**

