## [Peer Review File · Nature Communications]

Cross-property deep transfer learning framework for enhanced predictive analytics on small materials dataREVIEWER COMMENTS

Reviewer #1 (Remarks to the Author):

Comments on paper titled "Cross-property deep transfer learning framework for enhanced predictive analytics on small materials data" by Gupta et al. Submitted to Nature Communications

The authors present an interesting and timely study that proposes the use of a type of deep learning-based transfer learning method which seeks to use learned knowledge of properties stemming from large materials databases and applying the learned features to predict other properties not used in the initial training, properties for which comparatively small amounts of data are available.

This seems to be a fairly characteristic example of using transfer learning and have it be effective at predicting the property of interest. It reminds one of how transfer learning is used in object detection deep learning models (e.g. train on images of stop signs, detect images with speed limit signs). In this way, the strategy is not novel, however the strategy seems to be effective for the examples shown here and thus is likely useful, particularly for small datasets.

If only raw elemental fractions are used as input, how does one distinguish between different structure types or polymorphs for the same composition effectively? Two materials can have the same composition but different structure and thus quite different values of a property of interest.

The authors state that the overlap compositions are removed to avoid redundancy, which is good. However, how does one discern between materials that have the same composition but different structure, e.g. polymorphs of Al_2O_3 ? Was structural information also present in the databases and used as a guide for cleaning the data? Some description of how this was handled would be helpful.

Regarding the discussion of improving ElemNet between tensorflow versions: is the reduction in MAE (from 0.0417 eV/atom to 0.0405 eV/atom) calculated using the exact same train/validation split? Even if so, could these slight differences in MAE stem from the stochasticity of the model training process? This is not a big issue, but one wonders if the slight improvement in MAE is statistically significant or not.

The provided link for the JARVIS Figshare data appears to be broken.

In the SI, Table 5, the MAE for the bandgap energy is listed as 0.04 eV. Is this correct? It seems extremely low, but could be reasonable if these are training data errors. Are these listed errors the errors on training data or validation data?

The main results figures, Figures 2-4, are very difficult to parse visually, as they are essentially massive tables with many abbreviations, colors, and font styles. I think the authors need to find a way to visually represent their results with data plots that better highlight the key results. These table-style figures can still provide lots of information, and thus may be better suited to be moved to the SI.

The authors state that the feature extraction method from the pre-trained DL model is the best method with regard to getting TL-based results on new properties of small datasets. However, it is unclear how this feature extraction method is performed. Are feature vectors from the individual ElemNet layers somehow extracted and then used as new features to train the ML models e.g. AdaBoost? This sounds like the case from the Methods section. However, the features extracted from layers of different sizes (i.e. numbers of neurons) will have different shapes, correct? For instance, some feature vectors may have length 1024 and others 512. Therefore, when put into other ML algorithms, the feature matrix will not be a uniform size. How is this issue handled, or am I just confused?

In the results for Table 1, since 10-fold cross validation was performed, the authors should put error bars on the predicted formation enthalpy and bandgap values. In the case of bandgaps, the TL:DL result of 0.398 eV is very close to the AutoML(PA) result of 0.402 eV. The uncertainties from

cross validation are almost certainly larger than this very small 0.004 eV difference, so these results are likely within the error bars of the prediction. Some more discussion of the model uncertainty and how that plays into the choice of best model type for these properties would be helpful. One may also take into account model ease of use: AutoML with magpie-type features is likely easier to use than the transfer learning deep learning technique described here, so for the case of bandgaps one may wish to opt for the AutoML method as the errors are essentially degenerate and the non-deep learning method is likely easier and faster to use in practice.

The discussion section reads more like a summary/conclusion than an actual discussion of the results. Perhaps some content needs to be reorganized or relabeled accordingly.

The authors say that in the future they plan to develop web-based tools to deploy the best performing predictive models. They may wish to consider uploading their best models to DLHub (dlhub.org), which is actively being developed precisely for this purpose.

The deep learning methods used in this work use elemental fraction vectors as the input, which is great for simplicity. Some of the most state-of-the-art deep learning models for predicting materials properties use graph-based methods and materials structure as input. These methods should also be highly amenable to transfer learning, though may be slightly more difficult to use as some knowledge of material structure is needed. Can the authors discuss a bit more on the presently used methods and the possible interplay/relationship with the graph-based based deep learning methods?

Reviewer #2 (Remarks to the Author):

This paper is an empirical exploration of a large number of model variants exploring transfer learning in material properties. The fundamental question is interesting and builds on the author's previous work.

Unfortunately, the analysis method here is flawed in a way that makes all the conclusions and discussion unreliable. The fundamental issue is that in a great many places, the best performing models of a set of N models are compared to a single model (or a set of models of different size). Consider the columns of Table 2:

SC: ML(EF) is the min of the test set across 12 models, while TL: DL (FeatExtr) is the min of the test set across 16 models.

Especially for the DL models which will observe significant variation in performance even across training runs, this results in a statistically unfair comparison. It's like comparing test scores of a class of students by comparing a random student from one class with the best score of the students from another class.

It would also greatly strengthen the analysis if an estimate of the inherent variability of the estimates of model performance. For example, the author bold .811 vs. .821 but there is no data shown to the reader to say that is even a measurable difference in performance. I would suggest looking at either the variation of multiple trainings of the same models or from bootstraps of the test set to better understand what magnitude of difference in performance could actually be measured.

While the above is the most important issue, I have a few other issues with the paper.

Novelty: There is nothing novel with the fundamental idea here. Transfer learning is well established, even in materials with paper such as this:

Chen, Chi, Weike Ye, Yunxing Zuo, Chen Zheng, and Shyue Ping Ong. 2019. "Graph Networks as a Universal Machine Learning Framework for Molecules and Crystals." *Chemistry of Materials: A Publication of the American Chemical Society* 31 (9): 3564–72.

There are some novel ideas on some technical details of how the transfer learning is done.

However “a novel cross-property deep transfer learning framework” strains the author’s credibility.

As the author’s note, FineTune is a well established technique. They propose ModFineTune and put the experimental results in the paper. However, there is no analysis that ModFineTune is useful or even different from FineTune (by eye, I bet they are indistinguishable). If the author’s would like to introduce a new technique, they should actually evaluate it.

The author’s also miss a real opportunity. The large difference between TL:Freezing and TL:FineTune is actually quite interesting and a better understanding and explanation of that difference could make for an interesting scientific result.

The analysis connected to Figure 6 is quite incomplete. The visualization makes it difficult to actually decide what a real pattern is and what is us fooling ourselves. If a claim like “those material properties that are correlated with each other tend to perform better...” is made, a much better analysis needs to support that.

Lines 198-199 compare TL and SC. The claim is written to make TL seem positive, but this is not a rigorous test. From a quick sign test on the results, I get $p \approx .1$. I would interpret this as “neither SC or TL outperform each other”. From a practical perspective, the SC(PA) models are much more appealing than the complexity of the TL models. Therefore, this section tells me: don’t bother with these TL models because it’s a coin flip which one will work better. If the authors would like to make a different claim, they should support it.

A few last more minor points

“Thus preventing any possibility of model overfitting or confounding” is simply a false, hyperbolic statement. Being careful in your test set split does not “prevent” overfitting or all kinds of confounding

Your color bar scales on Figures 2 and 3 are mislabeled

Lines 272-273: Similar to the main concern above, I can’t tell from the words if you took the model with the best cross-validation score and reported its value on the holdout set or if you took the model with the best performance on the holdout set.

Lines 391-392: You need to report the details of your hyperparameter search rather than just saying it is “extensive”.

Reviewer #3 (Remarks to the Author):

The authors present a transfer learning (TL) framework that was developed to transfer knowledge from a deep learning (DL) model trained on a large dataset labeled with a single property to a small dataset containing multiple properties. The authors demonstrate their cross-property deep TL framework at the example of the materials databases OQMD (six properties considered, one of which is used for training at a time; source of the large dataset), and JARVIS (39 properties; source of the small dataset). To realize a truly data-driven learning process, the authors employ “elemental fractions” as independent variables (features), which do not contain any physical domain knowledge.

The authors find that their TL approach outperforms popular machine learning (ML)/DL models trained from scratch (i.e., without consideration of the OQMD database) for up to 37 out of 39 properties. Even if physical domain knowledge is incorporated into the scratch models, the TL approach by the authors yields superior predictions in 24 out of 39 cases. These results suggest that cross-property deep TL bears the potential to support researchers in situations where large datasets are unavailable for certain properties, which I would consider the rule rather than the exception. Therefore, the work by Agrawal and co-workers clearly has high potential significance for materials science and related fields (such as molecular and pharmaceutical science).

The initial step of the method proposed consists in the application of DL to a large dataset (OQMD, about 340K entries). A single property (six in total: formation enthalpy per atom, bandgap, total energy per atom, stability, magnetic moment, volume per atom) is selected to train the DL model,

which is a 17-layer neural network called ElemNet and is based on elemental fractions. In the manuscript, this DL model is referred to as the source model. After training, the source model or information extracted thereof is utilized to learn 39 properties of a smaller dataset (JARVIS, 557 to 28K entries, depending on the property) containing comparable systems. Those systems that overlap with OQMD were removed from training the source model to avoid bias. One of two possible TL paths can be chosen:

Path A (Fine-Tuning). The source model is retrained on JARVIS.

Path B (FeatExtr). Features of the source model are extracted and combined with other machine learning (ML)/DL models. This path was shown to be superior to path A in terms of prediction accuracy. It also allows users to combine the cross-property deep TL framework of the authors with a learning model of their choice.

The authors compare the performance of both TL paths against both the above-mentioned scratch models and a baseline (average property value over all training data). The latter is inferior to both TL and scratch ML/DL for all properties.

As much as I am confident that the current study is relevant, comprehensible, and reproducible (the authors made the production code openly available), it lacks — in my opinion — robustness of the conclusions drawn:

i) The authors show that TL generally outperforms scratch ML/DL with respect to predicting JARVIS properties. However, 90% of the JARVIS data are used for training (81%) and validation (9%), leaving only 10% of the data for testing purposes. Given that 90% of the dataset still corresponds to up to about 25K entries, the results reported by the authors are only informative to researchers who study systems and properties for which such rich data pool is available. I would argue that this scenario is the exception rather than the rule. To address a larger target audience, learning curves that measure prediction accuracy as a function of the training set size might be helpful.

ii) It is interesting to see that transfer learning across properties seems to work, but how fast does transferability decay? If the materials of the test set are very similar to those of the training set, the conclusions drawn by the authors may be too optimistic. A visualization of the dataset (separated into training[validation]/test) might be helpful. I am thinking of a two-dimensional data distribution in which the data points of the test set are colored according to the absolute error between prediction and actual property value. How fast does transferability decay / the error increase as a function of distance to the training data? How do these plots differ for TL and scratch ML/DL? Can one observe different decay rates for these two modes of learning?

iii) The relative performance of TL is, on average, better by $12 \pm 7\%$ compared to scratch ML/DL. This value refers to the mean absolute error (MAE), the only performance measure reported by the authors. It has been shown by Pernot and Savin (<https://aip.scitation.org/doi/10.1063/1.5016248>) that the MAE is an ambiguous measure of performance. They propose that a series of performance measure should be reported to increase the reliability of conclusions drawn from statistical analyses (<https://iopscience.iop.org/article/10.1088/2632-2153/aba184>). In particular, Pernot and Savin show that the 95% percentile of the distribution of absolute errors (Q95) is a less ambiguous measure of performance. In my opinion, the inclusion of additional performance measures (following the recommendations by Pernot and Savin) would most certainly corroborate the robustness of the authors' conclusions.

Furthermore, I find the manuscript lengthy considering that the discussion is very descriptive rather than explanatory. In my opinion, this applies particularly to the later sections "Source Model Based Analysis" and "Correlation Among Materials Properties". I think that the manuscript would benefit from removing these sections as they do not offer novel insight and rather distract the reader from the actual research problem. Regarding my previous comment ("... descriptive rather than explanatory"): although I am aware of the difficulty to gain insight into neural networks, it seems to be an obviously interesting question why transfer learning across properties works. For instance, is there a link between the transfer performance from a source property to a target property and the correlation between these properties?

Eventually, it appears to me that the list of references is not up to date and somewhat arbitrary, at least with regard to the following two sentences:

1) "The field of materials science has seen a growing application of Artificial Intelligence (AI) and Machine Learning (ML) techniques, which has significantly contributed to enhanced property prediction models as well as accelerated materials exploration and discovery 1–12." Several important contributions are missing, for instance (reviews/perspectives only):

- <https://www.nature.com/articles/s41563-020-0777-6>

- <https://doi.org/10.1021/acs.accounts.0c00785>

- <http://arxiv.org/abs/2102.08435>

- <https://www.nature.com/articles/s41578-018-0005-z>

- <https://www.nature.com/articles/s41586-018-0337-2>

- <http://science.sciencemag.org/content/361/6400/360>

2) "Therefore, Transfer Learning (TL) 33 is often applied to tackle limited dataset problems by utilizing the rich features extracted from large datasets 34–44." I am wondering that no references from 2020 and 2021 are cited. However, as I am not familiar with this particular branch of literature, it is difficult for me to assess whether this list is up to date or not.

Despite these points of criticism, I appreciate the very thorough and transparent methodological work presented by the authors. Therefore, I am confident that the manuscript would already benefit from a high-level revision addressing the analysis of results and the conclusions drawn thereof, without need for further computational work.

I wish the authors good luck.

Response to the Reviewers

We thank the reviewers for their critical assessment of our work. In the following we address their concerns point by point.

Reviewer 1

Comments on paper titled “Cross-property deep transfer learning framework for enhanced predictive analytics on small materials data” by Gupta et al. Submitted to Nature Communications

The authors present an interesting and timely study that proposes the use of a type of deep learning-based transfer learning method which seeks to use learned knowledge of properties stemming from large materials databases and applying the learned features to predict other properties not used in the initial training, properties for which comparatively small amounts of data are available.

This seems to be a fairly characteristic example of using transfer learning and have it be effective at predicting the property of interest. It reminds one of how transfer learning is used in object detection deep learning models (e.g. train on images of stop signs, detect images with speed limit signs). In this way, the strategy is not novel, however the strategy seems to be effective for the examples shown here and thus is likely useful, particularly for small datasets.

Q1.1 *If only raw elemental fractions are used as input, how does one distinguish between different structure types or polymorphs for the same composition effectively? Two materials can have the same composition but different structure and thus quite different values of a property of interest.*

Reply: This is a materials representation question. As the reviewer might be aware, there are two broad types of representations used for this problem: composition-based and structure-based, both of which have their advantages and limitations. By definition, the representations based on composition alone are unable to distinguish between different structure polymorphs of the same composition, which would end up being duplicates in the data, and thus would need to be removed before ML modeling. However, the resulting composition-based models could be used to make predictions without the need for structure as an input, which is useful since structure information is often unavailable or very expensive to calculate for new materials. Composition-based models can thus be used effectively for virtual combinatorial screening to narrow down the vast space of promising materials systems by identifying promising compositions, which could then be further explored with methods for structure prediction and structure-based property prediction models. In this work, we use composition-based representation, similar to [1, 2, 3, 4, 5, 6]. To deal with duplicates arising due to different structures of same composition, we only keep the most stable structure available in the database, i.e., the one with minimum formation energy, in line with previous works. We have added text in Results section (Datasets subsection) in the revised manuscript to better clarify this.

Q1.2 *The authors state that the overlap compositions are removed to avoid redundancy, which is good. However, how does one discern between materials that have the same composition but different structure, e.g. polymorphs of Al₂O₃? Was structural information also present in the databases and used as a guide for cleaning the data? Some description of how this was handled would be helpful.*

Reply: There are two preprocessing steps to remove duplicate/overlapping compositions. First, we deduplicate both the source and target datasets independently by keeping only the most stable structure for each unique composition, as explained in the response to the previous question. Next, we remove the overlapping compositions between the source and target dataset by removing them from the source dataset (New source dataset = Old source dataset - Target dataset). We have added text in Results section (Datasets subsection) in the revised manuscript to better clarify this.

Q1.3 *Regarding the discussion of improving ElemNet between tensorflow versions: is the reduction in MAE (from 0.0417 eV/atom to 0.0405 eV/atom) calculated using the exact same train/validation split? Even if so, could these slight differences in MAE stem from the stochasticity of the model training process? This is not a big issue, but one wonders if the slight improvement in MAE is statistically significant or not.*

Reply: We would like to thank the reviewer for this interesting comment. Yes, the exact same train/validation split was used for different Tensorflow versions. In order to find if the improvement is significant or not, we have now performed five runs with different train/validation splits each for TF1 and TF2 (the same set of five splits were used for both versions). An average MAE and standard deviation of 0.0393 ± 0.0008 was obtained for TF2 and 0.0415 ± 0.0002 for TF1, with TF2 giving a more accurate model than TF1 in all five cases. Statistical testing gave a p-value of 0.0016 using corrected paired Student’s t-test proposed by Nadeau and Bengio [7] suggesting that the improvement is statistically significant. There are also several comparative studies on open source deep learning frameworks [8, 9] that show the difference in performance for different deep learning frameworks.

Q 1.4 *The provided link for the JARVIS Figshare data appears to be broken.*

Reply: The provided link seems to work fine for us. In order to avoid any URL formatting issues, the reviewer may want to try copy-pasting it directly in the browser. Here it is again:
<https://ndownloader.figshare.com/files/22471022>

Q 1.5 *In the SI, Table S5, the MAE for the bandgap energy is listed as 0.04 eV. Is this correct? It seems extremely low, but could be reasonable if these are training data errors. Are these listed errors the errors on training data or validation data?*

Reply: The results mentioned in SI old Table S5 (now Table S 6) are the errors on validation data. A similar error for OQMD bandgaps was obtained in [2, 10, 11, 12]. The difference in bandgap values between OQMD and other DFT databases is well-recognized [13].

Q 1.6 *The main results figures, Figures 2-4, are very difficult to parse visually, as they are essentially massive tables with many abbreviations, colors, and font styles. I think the authors need to find a way to visually represent their results with data plots that better highlight the key results. These table-style figures can still provide lots of information, and thus may be better suited to be moved to the SI.*

Reply: We thank the reviewer for the suggestion. We have moved old Figures 2-4 (now Tables S 5, S 7 and S 9) to the SI as suggested. Moreover, we have modified these tables by replacing the relative MAEs to Actual MAE (for Table S 5) and mean and standard deviation of MAE from the 10-fold cross-validation (for Tables S 7 and S 9) of the best-performing model under each category to show the actual performance. This was done to aid in more rigorous statistical testing as suggested by Reviewer 2. In the main paper, we have added Tables 2-4 which show the actual MAE on the test set of the best performing model for SC and TL (identified using validation MAEs).

Q 1.7 *The authors state that the feature extraction method from the pre-trained DL model is the best method with regard to getting TL-based results on new properties of small datasets. However, it is unclear how this feature extraction method is performed. Are feature vectors from the individual ElemNet layers somehow extracted and then used as new features to train the ML models e.g. AdaBoost? This sounds like the case from the Methods section. However, the features extracted from layers of different sizes (i.e. numbers of neurons) will have different shapes, correct? For instance, some feature vectors may have length 1024 and others 512. Therefore, when put into other ML algorithms, the feature matrix will not be a uniform size. How is this issue handled, or am I just confused?*

Reply: Yes, the feature vectors from the individual ElemNet layers are extracted using an in-built library (the method for which is also provided in our github repository <https://github.com/GuptaVishu2002/ElemNet2.0>) and then used as new features to train the ML/DL models. We use the pre-trained model’s activations from each of the 16 layers of ElemNet as 16 possible materials representations for each compound in the target dataset to train the model i.e. each set of feature vector (1024, 512, ...) is used separately to train the ML/DL models which makes the input sizes uniform each time we perform model training. Hence, for each target materials property, we perform two runs for Fine-tuning (TL:FineTune and TL:ModFineTune), 16 runs TL:DL(FeatExtr), and 183 runs each for SC:ML(EF), SC:ML(PA) and TL:ML(FeatExtr). For TL models, this is done for each source model. We have added additional information in methods section to better clarify this.

Q1.8 *In the results for Table 1, since 10-fold cross validation was performed, the authors should put error bars on the predicted formation energy and bandgap values. In the case of bandgaps, the TL:DL result of 0.398 eV is very close to the AutoML(PA) result of 0.402 eV. The uncertainties from cross validation are almost certainly larger than this very small 0.004 eV difference, so these results are likely within the error bars of the prediction. Some more discussion of the model uncertainty and how that plays into the choice of best model type for these properties would be helpful. One may also take into account model ease of use: AutoML with magpie-type features is likely easier to use than the transfer learning deep learning technique described here, so for the case of bandgaps one may wish to opt for the AutoML method as the errors are essentially degenerate and the non-deep learning method is likely easier and faster to use in practice.*

Reply: We thank the reviewer for the suggestion. We have modified our results for ML models (including AutoML) to make the ML and DL models' evaluation more consistent in terms of using the exact same training/validation/testing splits, after going over the comments given by Reviewer 2. Further, in order to evaluate statistical significance of the accuracy differences between SC and TL models as suggested by Reviewer 2, we now perform cross-validation (CV) for all models in the multi-source TL analysis, including SC models used for comparison. For the 39 DFT-computed properties, we perform 10-fold CV (Tables S 7 and S 9 in the revised SI) to find the model with the best validation error, which is then used on the test set (Tables 3-4 in the revised paper). Since there are only two experimental datasets, we perform a more rigorous 100-fold CV so as to get more number of MAE observations for more accurate statistical testing (Table S 11 in the revised SI) to find the model with the best validation error, which is then used on the test set (Table 5 in the revised paper). Finally, for the experimental bandgap dataset, we have also performed TL using formation energy as the source model, following the general cross-property TL insight obtained from this study that of all the source properties studied in this work, formation energy performed the best for most of the target properties.

From the new results in Tables S 11 and 5, we find that the best TL model is significantly better than the best SC model for both experimental properties. It is also worth noting that we are only using ElemNet for our pre-trained/deep learning model and transfer learning without any optimization or hyper-parameter tuning. In contrast, AutoML uses a wide variety of algorithms and optimizations to find the best model. Despite this, the current results and the statistical significance analysis indicate a clear advantage of using the proposed cross-property TL approach in terms of accuracy improvement, which is expected to encourage practitioners to use this approach over existing approaches. Moreover, we provide all the necessary code to perform cross-property TL, including feature extraction and fine-tuning, which is expected to facilitate the adoption and further building upon the proposed techniques.

Q1.9 *The discussion section reads more like a summary/conclusion than an actual discussion of the results. Perhaps some content needs to be reorganized or relabeled accordingly.*

Reply: Thank you for your valuable suggestion. We have reorganized the figures of the later sub-sections of the Results section into Supplementary Information and added the explanation in the discussion section as also suggested by Reviewer 3.

Q1.10 *The authors say that in the future they plan to develop web-based tools to deploy the best performing predictive models. They may wish to consider uploading their best models to DLHub (dlhub.org), which is actively being developed precisely for this purpose.*

Reply: Thank you for your valuable suggestion. We will definitely consider uploading the best models to DLHub in the future.

Q1.11 *The deep learning methods used in this work use elemental fraction vectors as the input, which is great for simplicity. Some of the most state-of-the-art deep learning models for predicting materials properties use graph-based methods and materials structure as input. These methods should also be highly amenable to transfer learning, though may be slightly more difficult to use as some knowledge of material structure is needed. Can the authors discuss a bit more on the presently used methods and the possible interplay/relationship with the graph-based based deep learning methods?*

Reply: Thank you for your valuable suggestion. We have added some text in the Discussion Section along these lines.

Reviewer 2

This paper is an empirical exploration of a large number of model variants exploring transfer learning in material properties. The fundamental question is interesting and builds on the author’s previous work.

Q2.1 *Unfortunately, the analysis method here is flawed in a way that makes all the conclusions and discussion unreliable. The fundamental issue is that in a great many places, the best performing models of a set of N models are compared to a single model (or a set of models of different size). Consider the columns of Table 2: SC: ML(EF) is the min of the test set across 12 models, while TL: DL (FeatExtr) is the min of the test set across 16 models. Especially for the DL models which will observe significant variation in performance even across training runs, this results in a statistically unfair comparison. It’s like comparing test scores of a class of students by comparing a random student from one class with the best score of the students from another class.*

Reply: We thank the reviewer for this important comment. We should have made it more clear that we perform extensive hyperparameter search for the ML results. Table S 4 in the revised SI lists the hyperparameters explored for each of the 12 ML algorithms, leading to an exploration of 183 models for SC:ML(EF). On the other hand, for TL:DL(FeatExtr), the best model was selected from 16 models by using *only* ElemNet as the deep learning model architecture *without* any hyperparameter tuning for each of the materials properties. Moreover, the reviewer’s comment has helped us discover that for SC:ML(EF), all 183 models were run on the *test set*, of which the minimum test MAE was reported, whereas for TL:DL(FeatExtr), the 16 models were evaluated on the *validation set*, and the model with the best validation MAE was subsequently used on the test set to report the MAE. Hence, if at all there was any unfair advantage, it was in favor of ML and not DL. This happened because for ML models, we performed model selection using the testing set instead of a validation set, which was inconsistent from how we built and evaluated the DL models.

In order to fix this, we have made the following modifications in our workflow: i) we now use identical training/validation/testing splits for all models in multi-target analysis (Table S 5 in SI and Table 2 in main paper); ii) for multi-source analysis, we also do cross-validation for model selection using the training+validation set for all models (Tables S 7, S 9 and S 11 in SI show the mean and stddev of validation MAEs); iii) The best model selected based on validation MAE is used on the test set to obtain the test MAEs (Tables 3, 4 and 5 in main paper).

Further, in the new results, we have ensured that model selection is always done using validation MAEs and never using test MAEs, i.e., when comparing two models (or groups of models), the test set is looked at only once for each model (or group of models). Thus, in Tables 2-5, the “Best SC” and “Best TL” MAE values are not the minimum of a set of test MAEs. Rather, each of them is a single test MAE of a model selected based on a set of validation MAEs. Thus the new results should be statistically comparable. We again thank the reviewer for their help with this.

Q2.2 *It would also greatly strengthen the analysis if an estimate of the inherent variability of the estimates of model performance. For example, the author bold .811 vs. .821 but there is no data shown to the reader to say that is even a measurable difference in performance. I would suggest looking at either the variation of multiple trainings of the same models or from bootstraps of the test set to better understand what magnitude of difference in performance could actually be measured.*

Reply: We thank the reviewer for the suggestion. We have modified our workflow as described in response to the previous comment by incorporating cross-validation (CV) for all models in multi-source analysis, which now allows us to do statistical significance testing across models. Note that in order to keep the number of TL models tractable, we performed cross-validation only for multi-source analysis with the best TL modeling configuration for each source property identified using multi-target analysis, which was done with a single fold of training-validation split (Table S 5 in revised SI shows the same for one source property).

For the 39 DFT-computed properties, we perform 10-fold CV (Tables S 7 and S 9 in the revised SI). Since

there are only two experimental datasets, we perform a more rigorous 100-fold CV so as to get more number of MAE observations for more accurate statistical testing (Table S 11 in the revised SI). For each target property, the model with the best validation MAE is identified, including those models whose validation MAEs were not statistically distinguishable from that of the best model at $\alpha=0.05$, i.e., their p-value $> \alpha$. The p-values were calculated using corrected paired Student’s t-test proposed by Nadeau and Bengio [7].

Q 2.3 *Novelty: There is nothing novel with the fundamental idea here. Transfer learning is well established, even in materials with paper such as this: Chen, Chi, Weike Ye, Yunxing Zuo, Chen Zheng, and Shyue Ping Ong. 2019. “Graph Networks as a Universal Machine Learning Framework for Molecules and Crystals.” Chemistry of Materials: A Publication of the American Chemical Society 31 (9): 3564–72. There are some novel ideas on some technical details of how the transfer learning is done. However “a novel cross-property deep transfer learning framework” strains the author’s credibility.*

Reply: We thank the reviewer for recognizing that there is some novelty in this work. We agree that the fundamental idea of transfer learning has been around for many years, and would like to clarify that all we are proposing here is a “framework” for performing cross-property transfer learning in materials science which has not been done before. We have modified appropriate places in the paper to indicate that we are building upon the existing works on transfer learning. We have also added a citation to the above-mentioned work.

Q 2.4 *As the author’s note, FineTune is a well established technique. They propose ModFineTune and put the experimental results in the paper. However, there is no analysis that ModFineTune is useful or even different from FineTune (by eye, I bet they are indistinguishable). If the author’s would like to introduce a new technique, they should actually evaluate it.*

Reply: We thank the reviewer for raising the issue. ModFineTune is fundamentally similar to FineTune, except that in ModFineTune, the weights of the last layer are not transferred but are randomly reinitialized. We investigated ModFineTune because here we are performing *cross-property* transfer learning. The idea was that since the last layer would have learnt to predict the source property, it might become over-specific to the source property, which could possibly make fine-tuning the model for the target property more challenging in some cases. To see if there is a significant difference between FineTune and ModFineTune, we calculate the one-tailed p-value using Signed Test [14, 15] as we have their results on 39 different datasets. Here, the null hypothesis is “ModFineTune is not better than FineTune” and alternate hypothesis is “ModFineTune is better than FineTune”. After comparing the results from 6 (source properties) x 39 (target properties) = 234 cases, we get the p-value = 0.00104 using a sign test calculator [16], thereby rejecting the null hypothesis at $\alpha=0.05$, indicating that TL:ModFineTune performs significantly better as compared to TL:FineTune for cross-property transfer learning.

Q 2.5 *The author’s also miss a real opportunity. The large difference between TL:Freezing and TL:FineTune is actually quite interesting and a better understanding and explanation of that difference could make for an interesting scientific result.*

Reply: We thank the reviewer for the suggestion. We think that the large difference between TL:Freezing and TL:FineTune may be because the representation can change drastically after every layer, gradually becoming more specific to the particular property and dataset used for source model training, thus making it difficult for the target DL model to learn the target property, especially for cross-property TL. Moreover, this seems to be consistent with the fact that of the 16 layers of ElemNet, the representations extracted from only the first four layers were found to result in competitive models. We have added additional information in Discussion Section regarding this.

Q 2.6 *The analysis connected to Figure 6 is quite incomplete. The visualization makes it difficult to actually decide what a real pattern is and what is us fooling ourselves. If a claim like “those material properties that are correlated with each other tend to perform better...” is made, a much better analysis needs to support that.*

Reply: Figure 6 was mainly used to find a pair-wise relationship between different target properties and not any general pattern or trend from the heatmap. We have updated it by adding correlation values to make it easier to visualize and moved it to supplementary materials after reviewing the comments by Reviewer 1 and Reviewer 3. The insights gained from the heatmap are added to the Discussion section.

Q 2.7 *Lines 198-199 compare TL and SC. The claim is written to make TL seem positive, but this is not a rigorous test. From a quick sign test on the results, I get $p = .1$. I would interpret this as “neither SC or TL outperform each other”. From a practical perspective, the SC(PA) models are much more appealing than the complexity of the TL models. Therefore, this section tells me: don’t bother with these TL models because it’s a coin flip which one will work better. If the authors would like to make a different claim, they should support it.*

Reply: We thank the reviewer for raising the issue. As described earlier, we have modified our workflow by first performing one run for all the SC/TL models for multi-target analysis. We then identify the best-performing model for each category and perform 10-fold cross-validation for that model. Finally, the model which gives the best validation error among SC and TL models is used to perform model testing on the holdout test set. We have also added 95th percentile of the distribution of the absolute error in addition to MAE for the best performing models to increase the reliability of conclusions drawn from statistical analyses as suggested by reviewer 3.

As also mentioned earlier, we perform extensive hyperparameter search for the ML results. Table S 4 in the revised SI lists the hyperparameters explored for each of the 12 ML algorithms, leading to an exploration of 183 models. On the other hand, for DL models, we only use ElemNet (which was originally designed for formation energy and which uses only raw elemental fractions as input) as the deep learning model architecture without any hyperparameter optimization, and it still shows comparable or better performance compared to SC(PA) which uses different and more informative attributes as input, which we consider quite promising.

Furthermore, for the new results from the modified workflow, we have calculated the one-tailed p-value for test MAE comparison for 41 materials properties datasets (39 DFT-computed datasets and two experimental datasets) using Signed Test [14, 15] as we are dealing with different datasets. Here, the null hypothesis is “TL model is not better than SC model” and alternate hypothesis is “TL model is better than SC model”. If we take into account that the SC models are allowed PA as input, we get the p-value = 0.00397 for MAE comparison and p-value = 0.04291 for (Q_{95}) comparison using a sign test calculator [16], thus rejecting the null hypothesis at $\alpha=0.05$, suggesting that such difference in test MAE/ Q_{95} is unlikely to have arisen by chance. We can thus infer that in general TL models perform significantly better than SC models for cross-property transfer learning. We have added additional information in the Results section.

Q 2.8 *“Thus preventing any possibility of model overfitting or confounding” is simply a false, hyperbolic statement. Being careful in your test set split does not “prevent” overfitting or all kinds of confounding*

Reply: We thank the reviewer for catching that! Obviously the intent was to convey that ensuring no overlap between training and testing splits is important, without which the model can easily overfit. But we see how the previously used phrase could be misleading, and have now removed it.

Q 2.9 *Your color bar scales on Figures 2 and 3 are mislabeled*

Reply: Actually they were not mislabeled as they represented the values for the middle table in both the figures. We apologize for not explicitly mentioning that in the table description. Anyway, since we have performed the model training for ML models again (due to the previous inconsistency with their evaluation w.r.t. DL models as described before), we have changed the results from relative MAE to actual MAE, and performed statistical significance testing on validation MAEs. Since that allows us to identify top performing models for each target property in a more statistically rigorous way, we have now removed the color scale altogether in the revised manuscript.

Q 2.10 *Lines 272-273: Similar to the main concern above, I can’t tell from the words if you took the model with the best cross-validation score and reported its value on the holdout set or if you took the model with the best performance on the holdout set.*

Reply: We thank the reviewer for the comment. We have clarified it by modifying the text at the mentioned place. We took the model with the best cross-validation score and reported its value on the holdout set.

Q2.11 *Lines 391-392: You need to report the details of your hyperparameter search rather than just saying it is “extensive”.*

Reply: As mentioned earlier, we have now added a table with the list of algorithms and their respective hyper-parameters for performing the model training in the supplementary materials (Table S 4).

Reviewer 3

The authors present a transfer learning (TL) framework that was developed to transfer knowledge from a deep learning (DL) model trained on a large dataset labeled with a single property to a small dataset containing multiple properties. The authors demonstrate their cross-property deep TL framework at the example of the materials databases OQMD (six properties considered, one of which is used for training at a time; source of the large dataset), and JARVIS (39 properties; source of the small dataset). To realize a truly data-driven learning process, the authors employ “elemental fractions” as independent variables (features), which do not contain any physical domain knowledge.

The authors find that their TL approach outperforms popular machine learning (ML)/DL models trained from scratch (i.e., without consideration of the OQMD database) for up to 37 out of 39 properties. Even if physical domain knowledge is incorporated into the scratch models, the TL approach by the authors yields superior predictions in 24 out of 39 cases. These results suggest that cross-property deep TL bears the potential to support researchers in situations where large datasets are unavailable for certain properties, which I would consider the rule rather than the exception. Therefore, the work by Agrawal and co-workers clearly has high potential significance for materials science and related fields (such as molecular and pharmaceutical science).

The initial step of the method proposed consists in the application of DL to a large dataset (OQMD, about 340K entries). A single property (six in total: formation energy per atom, bandgap, total energy per atom, stability, magnetic moment, volume per atom) is selected to train the DL model, which is a 17-layer neural network called ElemNet and is based on elemental fractions. In the manuscript, this DL model is referred to as the source model. After training, the source model or information extracted thereof is utilized to learn 39 properties of a smaller dataset (JARVIS, 557 to 28K entries, depending on the property) containing comparable systems. Those systems that overlap with OQMD were removed from training the source model to avoid bias. One of two possible TL paths can be chosen:

Path A (Fine-Tuning). The source model is retrained on JARVIS.

Path B (FeatExtr). Features of the source model are extracted and combined with other machine learning (ML)/DL models. This path was shown to be superior to path A in terms of prediction accuracy. It also allows users to combine the cross-property deep TL framework of the authors with a learning model of their choice.

The authors compare the performance of both TL paths against both the above-mentioned scratch models and a baseline (average property value over all training data). The latter is inferior to both TL and scratch ML/DL for all properties.

As much as I am confident that the current study is relevant, comprehensible, and reproducible (the authors made the production code openly available), it lacks — in my opinion — robustness of the conclusions drawn:

Q3.1 *The authors show that TL generally outperforms scratch ML/DL with respect to predicting JARVIS properties. However, 90% of the JARVIS data are used for training (81%) and validation (9%), leaving only 10% of the data for testing purposes. Given that 90% of the dataset still corresponds to up to about 25K entries, the results reported by the authors are only informative to researchers who study systems and properties for which such rich data pool is available. I would argue that this scenario is the exception rather than the rule. To address a larger target audience, learning curves that measure prediction accuracy as a function of the training set size might be helpful.*

Reply: We thank the reviewer for the valuable suggestion. As the reviewer has indicated in his summary of the work above, we perform model training on 39 different materials properties from DFT-computed database and two different materials properties from experimental dataset with different training data sizes ranging from 28K to 500 entries in this work. In addition, following the reviewer’s suggestion, we performed additional model training experiments for formation energy for different training data size using the same test set (10% of the total data size) to create a learning curve that shows prediction accuracy as a function of the training set size.

Figure R1: Training curve for prediction accuracy of target formation energy for different training data sizes on a fixed test set.

Figure R 1 shows that in general, TL models outperforms SC models for all the training sizes for formation energy prediction. We have also added this figure in the SI and referred to it in the discussion section.

Q3.2 *It is interesting to see that transfer learning across properties seems to work, but how fast does transferability decay? If the materials of the test set are very similar to those of the training set, the conclusions drawn by the authors may be too optimistic. A visualization of the dataset (separated into training/[validation]/test) might be helpful. I am thinking of a two-dimensional data distribution in which the data points of the test set are colored according to the absolute error between prediction and actual property value. How fast does transferability decay / the error increase as a function of distance to the training data? How do these plots differ for TL and scratch ML/DL? Can one observe different decay rates for these two modes of learning?*

Reply: We thank the reviewer for the valuable suggestion. We agree it would be an interesting and useful analysis. Unfortunately the specific plot suggestion made by the reviewer was not entirely clear to us, but taking cue from the comment, we have tried to perform a transferability decay study as described below for formation energy as an illustrative target property.

The data points corresponding to the bottom 10% of formation energy values were set aside as the “Extrapolation test set”. The remaining data was divided into into training, validation, and test split by using the `train_test_split` function from the Sklearn library. The obtained test split was called as “Interpolation test split”. The lower values for formation energy indicate a more stable compound, and it is desirable to have a model that can predict (and maybe even extrapolate) the lower values accurately.

Here the materials property value range of the training data is [-2.41005, 4.25274] and of testing data is [-2.4096, 3.31337] for the Interpolation test split and [-4.38553, -2.41013] for the Extrapolation test split. The scatter plot of the prediction error analysis is shown in Figure R 2.

Figure R 2 shows that the best TL model performs significantly better as compared to the best SC model. The SC model is only able to predict values closer to -2 eV/atom, which was the lowest property value in

Figure R2: Prediction error analysis for predicting formation energy in JARVIS dataset using best SC and best TL model.

the training data, whereas the TL model can accurately predict even lower values. Hence one can observe different decay rates for these two modes of learning where the TL model outperforms the SC model for formation energy as materials property.

Q3.3 *The relative performance of TL is, on average, better by 12+-7% compared to scratch ML/DL. This value refers to the mean absolute error (MAE), the only performance measure reported by the authors. It has been shown by Pernot and Savin (<https://aip.scitation.org/doi/10.1063/1.5016248>) that the MAE is an ambiguous measure of performance. They propose that a series of performance measure should be reported to increase the reliability of conclusions drawn from statistical analyses (<https://iopscience.iop.org/article/10.1088/2632-2153/aba184>). In particular, Pernot and Savin show that the 95% percentile of the distribution of absolute errors (Q95) is a less ambiguous measure of performance. In my opinion, the inclusion of additional performance measures (following the recommendations by Pernot and Savin) would most certainly corroborate the robustness of the authors' conclusions.*

Reply: We thank the reviewer for the valuable suggestion. We have added the 95th percentile of the distribution of the absolute error for the best performing models in table S 12 for the models used in Table 2-5 in the SI for all the target properties and added text in the discussion section about it.

Q3.4 *Furthermore, I find the manuscript lengthy considering that the discussion is very descriptive rather than explanatory. In my opinion, this applies particularly to the later sections "Source Model Based Analysis" and "Correlation Among Materials Properties". I think that the manuscript would benefit from removing these sections as they do not offer novel insight and rather distract the reader from the actual research problem. Regarding my previous comment ("... descriptive rather than explanatory"): although I am aware of the difficulty to gain insight into neural networks, it seems to be an obviously interesting question why transfer learning across properties works. For instance, is there a link between the transfer performance from a source property to a target property and the correlation between these properties?*

Reply: We thank the reviewer for the valuable suggestion. We have removed the text of later sub-sections "Source Model-Based Analysis" and "Correlation Among Materials Properties" Result sections and moved the Figures to the Supplementary Materials. We have also added an explanation regarding the "correlation between different target properties" into the Discussion (which was initially briefly discussed in the "Correlation Among Materials Properties"). We did not see a clear link between the transfer performance from a source property to the target property.

Q3.5 *"The field of materials science has seen a growing application of Artificial Intelligence (AI) and Machine Learning (ML) techniques, which has significantly contributed to enhanced property prediction models as well*

as accelerated materials exploration and discovery 1–12.” Several important contributions are missing, for instance (reviews/perspectives only):

- <https://www.nature.com/articles/s41563-020-0777-6>
- <https://doi.org/10.1021/acs.accounts.0c00785>
- <http://arxiv.org/abs/2102.08435>
- <https://www.nature.com/articles/s41578-018-0005-z>
- <https://www.nature.com/articles/s41586-018-0337-2>
- <http://science.sciencemag.org/content/361/6400/360>

Reply: We thank the reviewer for highlighting the lack of references for several important contributions and providing their valuable suggestions. We have added and cited the above mentioned papers.

Q 3.6 “Therefore, Transfer Learning (TL) 33 is often applied to tackle limited dataset problems by utilizing the rich features extracted from large datasets 34–44.” I am wondering that no references from 2020 and 2021 are cited. However, as I am not familiar with this particular branch of literature, it is difficult for me to assess whether this list is up to date or not.

Reply: We thank the reviewer for highlighting the lack of references for more recent transfer learning (TL) related work. We have added and cited more recent TL works.

References

- [1] Meredig, B. *et al.* Combinatorial screening for new materials in unconstrained composition space with machine learning. *Physical Review B* **89**, 094104 (2014).
- [2] Ward, L., Agrawal, A., Choudhary, A. & Wolverton, C. A General-Purpose Machine Learning Framework for Predicting Properties of Inorganic Materials. *npj Computational Materials* **2**, 16028 (2016). URL <http://dx.doi.org/10.1038/npjcompumats.2016.28.1606.09551>.
- [3] Jha, D. *et al.* ElemNet: Deep learning the chemistry of materials from only elemental composition. *Scientific reports* **8**, 17593 (2018).
- [4] Jha, D. *et al.* Enhancing materials property prediction by leveraging computational and experimental data using deep transfer learning. *Nature communications* **10**, 1–12 (2019).
- [5] Goodall, R. E. & Lee, A. A. Predicting materials properties without crystal structure: Deep representation learning from stoichiometry. *Nature communications* **11**, 1–9 (2020).
- [6] Wang, A. Y.-T., Kauwe, S. K., Murdock, R. J. & Sparks, T. D. Compositionally restricted attention-based network for materials property predictions. *npj Computational Materials* **7**, 1–10 (2021).
- [7] Nadeau, C. & Bengio, Y. Inference for the generalization error. *Machine learning* **52**, 239–281 (2003).
- [8] Shatnawi, A., Al-Bdour, G., Al-Qurran, R. & Al-Ayyoub, M. A comparative study of open source deep learning frameworks. In *2018 9th international conference on information and communication systems (icics)*, 72–77 (IEEE, 2018).
- [9] Elshawi, R., Wahab, A., Barnawi, A. & Sakr, S. Dlbench: a comprehensive experimental evaluation of deep learning frameworks. *Cluster Computing* 1–22 (2021).
- [10] Jha, D. *et al.* IRNet: A general purpose deep residual regression framework for materials discovery. In *25th ACM SIGKDD International Conference on Knowledge Discovery and Data Mining*, 2385–2393 (2019).
- [11] Kailkhura, B., Gallagher, B., Kim, S., Hiszpanski, A. & Han, T. Y.-J. Reliable and explainable machine-learning methods for accelerated material discovery. *npj Computational Materials* **5**, 1–9 (2019).
- [12] Choubisa, H. *et al.* Crystal site feature embedding enables exploration of large chemical spaces. *Matter* **3**, 433–448 (2020).

- [13] Hegde, V. I. *et al.* Reproducibility in high-throughput density functional theory: a comparison of aflow, materials project, and oqmd. *arXiv preprint arXiv:2007.01988* (2020).
- [14] Sheskin, D. J. *Handbook of parametric and nonparametric statistical procedures* (Chapman and Hall/CRC, 2003).
- [15] Salzberg, S. L. On comparing classifiers: Pitfalls to avoid and a recommended approach. *Data mining and knowledge discovery* **1**, 317–328 (1997).
- [16] Statistics, S. S. Sign test calculator. <https://www.socscistatistics.com/tests/signtest/default.aspx> (2018). Accessed = 2021-08-21.

REVIEWERS' COMMENTS

Reviewer #1 (Remarks to the Author):

Comments on paper titled "Cross-property deep transfer learning framework for enhanced predictive analytics on small materials data" by Gupta et al. Submitted to Nature Communications (R1 revision)

Regarding the author's response to my original comments which they label as Q1.1 and Q1.2 in their rebuttal:

This is a materials representation question. As the reviewer might be aware, there are two broad types of representations used for this problem: composition-based and structure-based, both of which have their advantages and limitations. By definition, the representations based on composition alone are unable to distinguish between different structure polymorphs of the same composition, which would end up being duplicates in the data, and thus would need to be removed before ML modeling. However, the resulting composition-based models could be used to make predictions without the need for structure as an input, which is useful since structure information is often unavailable or very expensive to calculate for new materials. Composition-based models can thus be used effectively for virtual combinatorial screening to narrow down the vast space of promising materials systems by identifying promising compositions, which could then be further explored with methods for structure prediction and structure-based property prediction models. In this work, we use composition-based representation, similar to [1, 2, 3, 4, 5, 6]. To deal with duplicates arising due to different structures of same composition, we only keep the most stable structure available in the database, i.e., the one with minimum formation energy, in line with previous works. We have added text in Results section (Datasets subsection) in the revised manuscript to better clarify this.

I understand the need to restrict the model to use just composition-based features. I think then the limitations of the model need to be stated more clearly then, in the interest of transparency and so that a property value of two structural polymorphs cannot be distinguished.

Regarding Q1.4, the Figshare link indeed works.

It is great that the authors have the source code and a notebook to make new predictions on Github. They should also include the fitted model weights on Github or maybe Figshare (if it is too large) so that others can make predictions without having to re-train the full deep learning model. I think this is essential for maximum transferability. The Figshare API makes it easy to pull in large files like saved DL network weights if Github can't host the file for you.

Other than the small above comments, the authors have adequately addressed my previous comments.

Reviewer #2 (Remarks to the Author):

I am happy to see revisions to the evaluation process to deal with the obvious flaws in the first version of the manuscript. I have a few follow-on points I will detail below.

However, as I said in the first review, I think the level of novelty and broad interest of this work is low. The few technical innovations would be better suited towards a smaller computational audience.

A few technical and presentation points that I hope are helpful:

* Q2.2: "corrected paired Student's t-test proposed by Nadeau and Bengio". I do not know what this phrase means. From the methods proposed (from Table 1 in their paper), the closest method would be "corrected resampled t"

* Q2.4: This analysis should be included in the paper itself.

* Table S7 and S11. The meaning of the +- is not defined

* Page 9, lines 136 - 144. This reflects a fundamental misunderstanding of Dropout. It was specifically proposed to improve the generalization error. The use of drop out during inference as an uncertainty technique is a later concept (which has very questionable value, see [1] for a good evaluation/discussion, though this is a long running debate). Simply disabling drop out at inference would cause especially ReLU networks to have quite uncalibrated output (because the average activation at a layer would go up). So it is extremely surprising if it would improve the MAE. There is a standard technique of rescaling the weights of activations to make inference a deterministic process, but your text does not say that you do this. I did not look at your code, but it is possible your code is doing this without your knowledge.

* Page 9, lines 151. This tiered model selection process is unusual. Assuming your text means that the multiple layers of validation testing still results in exactly one model being run on the test set for each category (and the test set is never mixed in), I think it is fair, but in general you would be better served by following a more standard process. You may be interested in [2] [3]

* Your results in Table 2, 3, and 5 would still benefit from showing whether the difference between values in rows is significant or not in some way. Understanding variation in MAE for the same trained model, or the MAE of predicting to mean, or a well designed statistical test would be helpful to the reader.

[1] Osband, Ian, Zheng Wen, Mohammad Asghari, Morteza Ibrahimi, Xiyuan Lu, and Benjamin Van Roy. 2021. "Epistemic Neural Networks." arXiv [cs.LG]. arXiv. <http://arxiv.org/abs/2107.08924>.

[2] Bates, Stephen, Trevor Hastie, and Robert Tibshirani. 2021. "Cross-Validation: What Does It Estimate and How Well Does It Do It?" arXiv [stat.ME]. <http://arxiv.org/abs/2104.00673>.

[3] <http://citeseerx.ist.psu.edu/viewdoc/summary?doi=10.1.1.47.6720>

Reviewer #3 (Remarks to the Author):

The authors addressed all remarks by the reviewers in a thorough, convincing, and constructive way. The manuscript's length reflects its information content. The line of reasoning is comprehensible and complete. In addition to my previous assessment (the study is relevant, comprehensible, and reproducible), the robustness of conclusions has increased substantially due to the revisions made by the authors.

Response to the Reviewers

We thank the reviewers for their thorough assessment of our work. In the following we address their remaining comments.

Reviewer 1

Comments on paper titled “Cross-property deep transfer learning framework for enhanced predictive analytics on small materials data” by Gupta et al. Submitted to Nature Communications (R1 revision)

Q1.1 *Regarding the author’s response to my original comments which they label as Q1.1 and Q1.2 in their rebuttal:*

“This is a materials representation question. As the reviewer might be aware, there are two broad types of representations used for this problem: composition-based and structure-based, both of which have their advantages and limitations. By definition, the representations based on composition alone are unable to distinguish between different structure polymorphs of the same composition, which would end up being duplicates in the data, and thus would need to be removed before ML modeling. However, the resulting composition-based models could be used to make predictions without the need for structure as an input, which is useful since structure information is often unavailable or very expensive to calculate for new materials. Composition-based models can thus be used effectively for virtual combinatorial screening to narrow down the vast space of promising materials systems by identifying promising compositions, which could then be further explored with methods for structure prediction and structure-based property prediction models. In this work, we use composition-based representation, similar to [1, 2, 3, 4, 5, 6]. To deal with duplicates arising due to different structures of same composition, we only keep the most stable structure available in the database, i.e., the one with minimum formation energy, in line with previous works. We have added text in Results section (Datasets subsection) in the revised manuscript to better clarify this.”

I understand the need to restrict the model to use just composition-based features. I think then the limitations of the model need to be stated more clearly then, in the interest of transparency and so that a property value of two structural polymorphs cannot be distinguished.

Reply: Thank you for your valuable suggestion. We have added text in the “Model Architectures” under Methods Section along these lines.

Q1.2 *Regarding Q1.4, the Figshare link indeed works. It is great that the authors have the source code and a notebook to make new predictions on Github. They should also include the fitted model weights on Github or maybe Figshare (if it is too large) so that others can make predictions without having to re-train the full deep learning model- I think this is essential for maximum transferability. The Figshare API makes it easy to pull in large files like saved DL network weights if Github can’t host the file for you.*

Reply: We thank the reviewer for the valuable feedback. We have provided the source models in the github under elemnet/model directory and the target models via a zenodo link which is also provided in the github.

Q1.3 *Other than the small above comments, the authors have adequately addressed my previous comments.*

Reply: We thank the reviewer for investing time into providing us with valuable feedback.

Reviewer 2

I am happy to see revisions to the evaluation process to deal with the obvious flaws in the first version of the manuscript. I have a few follow-on points I will detail below.

However, as I said in the first review, I think the level of novelty and broad interest of this work is low. The few technical innovations would be better suited towards a smaller computational audience.

A few technical and presentation points that I hope are helpful:

Q2.1 Q2.2: *“corrected paired Student’s t-test proposed by Nadeau and Bengio”. I do not know what this phrase means. From the methods proposed (from Table 1 in their paper), the closest method would be “corrected resampled t”*

Reply: We thank the reviewer for correcting the terminology. We have also added the term in the caption of Table S7, S9 and S11 with the correct terminology.

Q2.2 Q2.4: *This analysis should be included in the paper itself.*

Reply: Thank you for your valuable suggestion. We have modified text in Discussion Section along these lines.

Q2.3 Table S7 and S11. *The meaning of the +- is not defined*

Reply: Thank you for your feedback. We have modified the caption of Table S7, S9 and S11 to better explain the meaning of the +-.

Q2.4 Page 9, lines 136 - 144. *This reflects a fundamental misunderstanding of Dropout. It was specifically proposed to improve the generalization error. The use of drop out during inference as an uncertainty technique is a later concept (which has very questionable value, see [1] for a good evaluation/discussion, though this is a long running debate). Simply disabling drop out at inference would cause especially ReLU networks to have quite uncalibrated output (because the average activation at a layer would go up). So it is extremely surprising if it would improve the MAE. There is a standard technique of rescaling the weights of activations to make inference a deterministic process, but your text does not say that you do this. I did not look at your code, but it is possible your code is doing this without your knowledge.*

Reply: We thank the reviewer for the feedback. After carefully examining the code we found that actually dropout was disabled both during the training and inference phases. We have changed the text under Model Architecture Design to clearly specify this.

Q2.5 Page 9, lines 151. *This tiered model selection process is unusual. Assuming your text means that the multiple layers of validation testing still results in exactly one model being run on the test set for each category (and the test set is never mixed in), I think it is fair, but in general you would be better served by following a more standard process. You may be interested in [2] [3]*

Reply: We thank the reviewer for the valuable feedback in providing the relevant references. We will definitely look into it.

Q2.6 Your results in Table 2, 3, and 5 would still benefit from showing whether the difference between values in rows is significant or not in some way. Understanding variation in MAE for the same trained model, or the MAE of predicting to mean, or a well designed statistical test would be helpful to the reader.

Reply: We thank the reviewer for the valuable feedback. As suggested, we have added the test MAE obtained from the base model which uses the average property value of all the training data provided to it as the predicted property of a test compound i.e. the MAE of predicting the mean in Tables 2, 3, 4 and 5.

Reviewer 3

The authors addressed all remarks by the reviewers in a thorough, convincing, and constructive way. The manuscript’s length reflects its information content. The line of reasoning is comprehensible and complete. In addition to my previous assessment (the study is relevant, comprehensible, and reproducible), the robustness of conclusions has increased substantially due to the revisions made by the authors.

Reply: We thank the reviewer for their encouraging feedback! It is very much appreciated.